# Antibiotic Therapy Strategies for Treating Gram-Negative Severe Infections in the Critically Ill: A Narrative Review

**DOI:** 10.3390/antibiotics12081262

**Published:** 2023-07-31

**Authors:** Alberto Corona, Vincenzo De Santis, Andrea Agarossi, Anna Prete, Dario Cattaneo, Giacomina Tomasini, Graziella Bonetti, Andrea Patroni, Nicola Latronico

**Affiliations:** 1Accident, Emergency and ICU Department and Surgical Theatre, ASST Valcamonica, University of Brescia, 25043 Breno, Italy; 2AUSL Romagna, Umberto I Hospital, 48022 Lugo, Italy; 3Accident, Emergency and ICU Department, ASST Santi Paolo Carlo, 20142 Milan, Italy; 4Unit of Clinical Pharmacology, ASST Fatebenefratelli Sacco University Hospital, Via GB Grassi 74, 20157 Milan, Italy; 5Urgency and Emergency Surgery and Medicine Division ASST Valcamonica, 25123 Brescia, Italy; 6Clinical Pathology and Microbiology Laboratory, ASST Valcamonica, 25123 Brescia, Italy; 7Medical Directorate, Infection Control Unit, ASST Valcamonica, 25123 Brescia, Italy; 8Department of Medical and Surgical Specialties, Radiological Sciences and Public Health, University of Brescia, 25123 Brescia, Italy

**Keywords:** antibiotic therapy, Gram-negative, infection, critically ill patients, strategy, combination, monotherapy, resistance

## Abstract

Introduction: Not enough data exist to inform the optimal duration and type of antimicrobial therapy against GN infections in critically ill patients. Methods: Narrative review based on a literature search through PubMed and Cochrane using the following keywords: “multi-drug resistant (MDR)”, “extensively drug resistant (XDR)”, “pan-drug-resistant (PDR)”, “difficult-to-treat (DTR) Gram-negative infection,” “antibiotic duration therapy”, “antibiotic combination therapy” “antibiotic monotherapy” “Gram-negative bacteremia”, “Gram-negative pneumonia”, and “Gram-negative intra-abdominal infection”. Results: Current literature data suggest adopting longer (≥10–14 days) courses of synergistic combination therapy due to the high global prevalence of ESBL-producing (45–50%), MDR (35%), XDR (15–20%), PDR (5.9–6.2%), and carbapenemases (CP)/metallo-β-lactamases (MBL)-producing (12.5–20%) Gram-negative (GN) microorganisms (i.e., *Klebsiella pneumoniae, Pseudomonas aeruginosa,* and *Acinetobacter baumanii*). On the other hand, shorter courses (≤5–7 days) of monotherapy should be limited to treating infections caused by GN with higher (≥3 antibiotic classes) antibiotic susceptibility. A general approach should be based on (i) third or further generation cephalosporins ± quinolones/aminoglycosides in the case of MDR-GN; (ii) carbapenems ± fosfomycin/aminoglycosides for extended-spectrum β-lactamases (ESBLs); and (iii) the association of old drugs with new expanded-spectrum β-lactamase inhibitors for XDR, PDR, and CP microorganisms. Therapeutic drug monitoring (TDM) in combination with minimum inhibitory concentration (MIC), bactericidal vs. bacteriostatic antibiotics, and the presence of resistance risk predictors (linked to patient, antibiotic, and microorganism) should represent variables affecting the antimicrobial strategies for treating GN infections. Conclusions: Despite the strategies of therapy described in the results, clinicians must remember that all treatment decisions are dynamic, requiring frequent reassessments depending on both the clinical and microbiological responses of the patient.

## 1. Introduction

Early administration of targeted antibiotic therapy is the mainstay of treatment for severe infections; however, data informing the correct duration and type (mono or combination) of antibiotic regime are still missing. The duration and type of antibiotic therapy are generally based on the site of infection, severity of illness, and pathogen characteristics (i.e., inoculum size, virulence, susceptibility, and biofilm formation capacity) [1]. Long antibiotic courses have no additional benefit and may favor the development of antimicrobial resistance (AMR), whereas shorter ones may be a solution to reduce antibiotic-related complications and the development of antibiotic resistance due to lower antibiotic pressure. Multiple studies over the past two decades have evaluated different schemes of antibiotic therapy with the goal of achieving an equally effective, shorter regimen [2]. Reducing antibiotic overuse is an important target of an antibiotic stewardship intervention [3,4].

Recently, several retrospective and prospective series of Gram-negative (GN)-related infections have been published, analyzing the efficacy and impact of treatment duration and type on clinical outcome.

The aim of this narrative review is to explore the most relevant published literature data on the duration and type (combination or monotherapy) of antimicrobial therapy focusing on bacteremia (B), pneumonia, and intra-abdominal infection (IAI) caused by GN.

## 2. Results

### 2.1. Epidemiology of GN Microrganisms and Antibiotic Resistance

Nearly one-half of total B is caused by GN, and the mortality rate can reach 40% in severe cases [1,2]. The prevalence of nosocomial acquired (N-A) GN B ranges from 41–65% [5,6], with a variable attributable mortality of 6.9–60% [3,4,5,6,7,8], higher in *Klebsiella* spp., *Acinetobacter* spp. (60%), and *Pseudomonas* spp. (>40%) [5,6]. Over the last three decades, community-acquired C-A GN B episodes increased from about 63.5 episodes up to 141.9 per 100,000 inhabitants [7,8] and were related to elevated mortality (44.5–77.4%) [5,6,7,8]. The types of microorganisms and their resistance rates are shown in Table 1 and Table 2. A reliable definition of resistance patterns for the GN microorganisms seems to be [4]: (i) MDR: Non-susceptible to at least three agents in antimicrobial categories; (ii) XDR: Susceptible to only one antibiotic; and (iii) PDR: Non-susceptible to all antimicrobial agents listed. Indeed, CPE may be resistant to colistin (20–55.2%), tigecycline, aminoglycosides, quinolones, and cephalosporins [9,10]. An important impact on antibiotic choice is related to the type of carbapanemases produced by microorganisms.

*Ambler Class A* hydrolyzes a limited range of penicillins and is mainly found in Gram-positive bacteria. It degrades cephalosporins, extended-spectrum cephalosporins, monobactams, and BLIBL. In the late 1990s, class A β-lactamases underwent a mutation, becoming carbapenemases, able to degrade carbapenems. Six types of class A carbapenemases exist, of which the most representative is KPC, transmitted via a plasmid to generate CR Enterobacteriaceae and GN.

*Ambler Class C* β-lactamases are derived from the *ampC* gene located on the genome of many of the Enterobacteriaceae. They are cephalosporinases resistant to clavulanic acid but sensitive to cefoxitin and ceftazidime. Moreover, in large amounts, they may exhibit resistance to carbapenems.

*Ambler Class D,* known as OXA enzymes, which include OXA-1 and OXA-10, have an active serine site similar to class A and C β-lactamases, showing hydrolyzing activity against cloxacillin-, oxacillin-, and extended-spectrum cephalosporins too.

*Ambler Class B MBL* β-lactamases show the metal Zn^2+^ in the point of the enzyme’s active center. MBL degrades all β-lactam agents but monobactams. To date, carbapenem-resistant metallo-β-lactamases (IMP, VIM), SPM, GIM, NDM, and FIM have been reported.

### 2.2. Bacteraemia (B)

B is the growth of a microorganism in the blood cultures associated with clinical signs of infection; its severity of illness may range from asymptomatic to septic shock, and it is still correlated with high mortality (21–55%). B may be classified as (i) C-A if occurring <48 h after hospital admission; and (ii) N-A if not in incubation and occurring >48 h after the patient’s admission to hospital. B may be considered “*primary*” or “*secondary*” if correlated with no recognized source or a deep-seated infection, respectively. Up to date, no data exist to inform optimal antibiotic therapy. Particularly controversial data are reported for MDR, XDR, and PDR-GN microorganisms in terms of both type (mono- or combination of antimicrobial agents) and duration (short- and long-courses).

#### 2.2.1. Antibiotic Duration Therapy and Type in GN Bloodstream Infections

##### MDR and ESBL-p GN B and Treatment Options

ESBLs are a group of β-lactamases that hydrolyze third-generation cephalosporins and aztreonam but are inhibited by clavulanic acid and have evolved in Enterobacteriaceae since 1980 [11]. Many Enterobacteriaceae may harbor AmpC β-lactamases, cephalosporinases that inactivate most penicillins, BLIBL, and cephalosporins. As a result, ESBL-producing organisms pose major treatment problems because of their cross-resistance to most antimicrobial classes. Early carbapenems should be considered, particularly if associated with aminoglycosides since the bactericidal activity was found to be greater with the faster killing rates of amikacin, although no evidence may support their synergy [11]. Tigecycline, temocillin, and polymyxins are reserved for patients with microorganisms resistant to all of the classes of antibiotics. Lee et al. compared the efficacy of fluoroquinolones and carbapenems for ESBL B due to *Escherichia coli* and *Klebsiella pneumoniae,* and the former resulted in being non-inferior; conversely, a few studies have found higher efficacy in carbapenems [12,13]. The authors identified 103 patients undergoing empiric and targeted therapy based on BLBLI (n = 72) or carbapenems (n = 31). The mortality rate for patients undergoing empiric and targeted BLBLI was lower (5.9% vs. 9.4%) than that of those who switched to carbapenems and of those (16.7%) receiving empiric/definitive therapy with carbapenems. BLBLI may reduce the spread of *Klebsiella*-carrying resistance plasmids and may facilitate ESBL mutations to less harmful enzymes [14]. A study reporting a total of 331 patients with ESBL-B, of whom 48% received piperacillin-tazobactam and 52 carbapenems, showed that the risk of death was 1.92 times higher for the former group [15]. The largest experience with the efficacy of *Temocillin* is an observational study conducted in 92 patients caused by ESBL-producing Enterobacteriaceae with or without de-repressed AmpC β-lactamases [16]. The recorded clinical and bacteriological efficacies were 86% and 84%, respectively. Most recently, Matsumura et al. [17] evaluated 25 patients who were given cefmetazole or flomoxef and experienced a lower sepsis-related organ failure rate compared to 45 patients receiving meropenem. The efficacy of cefepime in the management of ESBL-producing Enterobacteriaceae remains to be assessed. In a multicenter [18] retrospective study considering patients undergoing cefepime or meropenem therapy for ESBL-p *E. coli* and *K. pneumoniae* B, the crude 30-day mortality was not statistically different if compared to carbapenems. Moreover, the 30-day mortality rates were lower (0%) in the case of isolates with a cefepime MIC ≤ 1 mcg/mL. A total of 78 patients with primary B due to AmpC-producing β-lactamase Enterobacteriaceae that were [11] given cefepime or meropenem showed a lower mortality rate (31.2 vs. 34.3%) in the cefepime group. Siedner et al. evaluated the use of cefepime compared to other antibiotics for Enterobacteriaceae treatment [11], reporting that patients receiving the former showed no difference in in-hospital mortality compared to those on carbapenems (17% vs. 26%). However, a greater number of cases with persistent B were recorded in the group receiving monotherapy with carbapenems (25% vs. 0%). The same results were found by Blanchette et al. [19]. In a retrospective cohort study of 144 patients with Enterobacteriaceae B, cefepime and carbapenems showed the same crude 30-day mortality rates (26.4% vs. 22.2%). However, in the case of an isolate with a cefepime MIC of ≥4 mcg/mL, significantly higher mortality was noticed, suggesting the use of cefepime only for isolates with MICs ≤ 2 mcg/mL.

##### XDR, PDR and CPR GN B and Treatment Options

The overuse of carbapenems for treating ESBL-p microorganisms caused the emergence of CP Enterobacteriaceae such as CP *Klebsiella pneumoniae*, whose prevalence ranges from <1% up to >50%. KPC is also observed in *Escherichia coli, Enterobacter cloacae,* and *Pseudomonas aeruginosa* [20]. CPEs exhibit an elevated MIC (0.12 mg/L to >256 mg/L) for carbapenems. To obtain a bactericidal effect for strains with MICs ≤ 4 mg/L, a prolonged infusion of high-dose carbapenems is needed to achieve sufficient free time above the MIC (i.e., >40%), although treatment of CPE infection with carbapenems alone should not be encouraged if MICs are >8 mg/liter [21,22].

The efficacy of monotherapy with carbapenems for CP *Klebsiella pneumoniae,* based on MIC [23], ranged from 69% (MIC ≤ 4 μg/mL) to only 29% (MIC > 8 μg/mL). The lowest mortality rate was found in patients undergoing (i) carbapenems’ combination therapy (MIC ≤ 4 μg/mL) or (ii) carbapenems’ regimes compared with non-carbapenems’ regimes (12% vs. 41%) [23]. In a recent review, the mortality rate of patients treated with monotherapy with carbapenems was indeed high (40.1%) [24].

*Polymixins.* A concern with the use of polymyxins is their resistance development during treatment. A monotherapy with only colistin was given (loading dose of 9 MU followed by a maintenance dose of 4.5 MU, twice daily) to 14 patients developing GN Enterobacteriaceae B caused by microorganisms susceptible only to colistin [25] and was associated with a clinical cure rate of 82.1% [26]. A prospective study on 258 patients treated with colistin showed that 21.7% of patients with the highest total daily dose (9 MU/day) died compared with 27.8% and 38.6% of patients on lower doses (6 and 3 MU/day) [27].

*Tigecycline*’s susceptibility is generally maintained in CRE (97.4%), and it should be reserved, doubling the posology (100 mg twice daily) [28], for cases without any other treatment options, considering that monotherapy does not correlate with a favorable outcome.

*Fosfomicin* has antimicrobial activity against KPC *Pseudomonas aeruginosa* and NDM-1-producing Enterobacteriaceae [28,29]. The most extensive study was in 48 critically ill ICU patients treated with fosfomycin for infections due [30] to fosfomycin-susceptible CPE infections, including mainly B (52.1%). Fosfomycin was given for a median of 14 days, mainly in combination with colistin, tigecycline, or gentamicin, and was associated with clinical and microbiological success in 54.2% and 56.3%, respectively [30]. Four major systematic reviews or meta-analyses have been published regarding treatment options for CR Enterobacteriaceae [24,31,32] even if most information regarding the antimicrobial regimen is derived from observational studies. Literature data suggest major efficacy rates when antibiotics are used in combination therapy (27.2–40%) [28].

A meta-analysis reports (i) 18 studies on mortality and (ii) two other studies on therapy failure in 651 patients undergoing targeted antibiotic treatment. Fifteen studies reported on CPE and five others on CR Enterobacteriaceae, and one study included infections due to CR *Klebsiella pneumoniae*. *Klebsiella* spp. was the only responsible pathogen in fourteen studies and the predominant one in five other studies. In 8 out of 20 studies, the total, or the majority, of the included infections were B. Mortality varied according to the combination of antibiotics: (i) Tigecycline-colistin: 0–30%; (ii) tigecycline-gentamicin: 0–50/64%; (iii) carbapenems-colistin: 0–67%; and (iv) colistin-gentamicin: 40–61/67%. Regarding patients who received monotherapy, mortality varied between carbapenems (9–50%), tigecycline (0–53/80%), colistin (33–57%), and gentamicin (6.3–80%). In the case of CR *Klebsiella pneumoniae*, the combination of tigecycline-colistin was related to a mortality of 25–31%, while monotherapy with carbapenems, colistin, and tigecycline was associated with a higher 30-day mortality (50–73%).

In another study including 10 patients, the recorded mortality was 50% for patients treated with amikacin in association with carbapenems. Combination therapy (40%) was associated with a higher percentage of success than monotherapy (16.7%). A study reporting mainly ICU patients showed patients undergoing either tigecycline-colistin or colistin-tigecycline-gentamicin combination therapy had, respectively, 42.9% and 0% failures [24].

Qureshi et al. compared the mortality of 15 patients receiving combination therapy with that of 19 patients on monotherapy (colistin or meropenem), which was 13.3% and 57.8%, respectively [33]. Tumbarello et al. conducted a prospective observational study on 125 patients with CP *Klebsiella pneumoniae* B and found 30-day mortality rates of 34.1% and 54.1%, respectively, for combination and monotherapy [22]. Daikos et al. published a study on 28-day mortality rates in 205 patients with CP *Klebsiella pneumoniae* B with a high (25.4%) rate of colistin resistance [21]. Survival was higher in those treated with combination therapy (44.4%) than in those undergoing monotherapy (27.2%). The beneficial effect of combination therapy was maximized in cases of rapidly fatal underlying diseases or septic shock [21]. A recent systematic review by Tzouvelekis [33] et al. presents the clinical outcome of CP Enterobacteriaceae infections in 889 patients, with 49% undergoing combination, 39% monotherapy, and 12% inappropriate therapy. Monotherapy was associated with a mortality rate of 40.1% for carbapenems, 41.1% for tigecycline, and 42.8% for colistin, while inappropriate therapy was associated with a mortality rate of 46.1%.

A meta-analysis examining 39 published studies on 1054 CPEn found a strategy of combination therapy applied for 77% of *Acinetobacter baumannii*, 44% of *Klebsiella pneumoniae*, and 50% of *Pseudomonas aeruginosa.* Its use led to less resistance development in vitro. High synergy rates support the combination of a carbapenem with a polymyxin against *Acinetobacter baumannii*. The efficacy of carbapenem combination therapy appears to be MIC-dependent too [31]. Moreover, the mortality rate related to combination therapy based on the use of meropenem for treating CP *Klebsiella pneumoniae* B has been increasing from 19.4% (for MIC ≤ 8 μg/mL) up to 35.5% (for MIC > 8 μg/mL) [28]. The rationale of the proposal to use two carbapenems has been based on the fact that ertapenem, having a high affinity for the CP *Klebsiella pneumoniae* enzyme, would serve as a decoy, allowing for the second carbapenem (meropenem or doripenem) to be protected from K *Klebsiella pneumoniae* [26,34]. Such associations showed clinical and microbiological success of 80.7% and 96%, respectively, in treating 17 septic patients with XDR or PDR CP *Klebsiella pneumoniae* B. A review article focusing on various combinations of colistin compared with monotherapy [35] concluded that the former approach offered a benefit to the survival of the patients. A systematic review analyzing PCRTs considering more than 1000 patients treated with polymyxin therapy for CPE infections [36] stated combination therapy was associated with lower 30-day mortality if compared to monotherapy. A narrative review of 15 studies that included 55 unique patients found that monotherapy based on colistin monotherapy was associated with lower (14.3% vs. 72.7%) clinical success than combination therapy for CP *Klebsiella pneumoniae* infections [37]. In a recent cohort study, colistin-based combination therapy was associated with better 28-day survival than non-colistin regimes (33.3% vs. 5.5%; *p* = 0.018) in treating 36 patients with B due to CR Enterobacteriaceae [38]. In cases of CP *Klebsiella pneumoniae* susceptible to aminoglycosides, gentamicin monotherapy or in combination with tigecycline may reduce mortality from sepsis caused by CP *Klebsiella pneumoniae* [39]. Two synergistic active drugs, that is, colistin, tigecycline, gentamicin, or carbapenems (when MIC < 4 μg/mL), were found superior if compared to their use in monotherapy to treat CP *Klebsiella pneumoniae*, B [40]. Another review found similar mortality results in terms of comparison between combination and monotherapy (18.3% vs. 49.1%) [41]. Many prospective observational studies have tried to establish the efficacy of the antibiotic combination strategy in the treatment of CP *Klebsiella pneumoniae*, B [24,28,42,43]. In one of the largest cohort studies to date, combination therapy was again associated with lower mortality than monotherapy (27.2% vs. 44.4%) [28]. Moreover, synergistic association therapy was an independent predictor of survival (mainly if based on the effectiveness of carbapenem-containing regimens), albeit the benefits of combination therapy may outweigh the risks and should be adopted particularly in the treatment of severe CR Enterobacteriaceae infections [24,28,31,34,37,38,43,44,45,46,47].

##### The New Antibiotics against XDR, PDR, DTR and CR GN

Ceftolozane-tazobactam is a combination of a sixth-generation cephalosporin combined with a β-lactamase inhibitor active against MDR and XDR *Pseudomonas aeruginosa,* and its action is based on inhibition of the key penicillin-binding proteins (PBPs) with an activity against ESBL-producing Enterobacteriaceae too [48,49,50], being inactive against neither carbapenemase-producing nor MBL [51,52]. The use of ceftolozane-tazobactam has been recommended as an efficacious option for DTR *Pseudomonas aeruginosa* infections because of its high susceptibility rates [53]. A recent European survey monitoring the in vitro activity of ceftolozane-tazobactam found *Pseudomonas aeruginosa* isolates susceptible in 94.1% of isolates coming from occidental Europe and 80.9% of isolates coming from oriental Europe. Susceptibility rates of 75.2% and 59.2% were found in occidental and oriental Europe, respectively, in the case of CRPA isolates [54], whereas 88.7% susceptibility was found in a partial cohort tested (71.2%) of *Pseudomonas aeruginosa* isolates (95.8% MDR and 37.7% XDR), representing 91.1% of the entire cohort of MDR GN infections in the US medical ICU. Published European data about resistant phenotypes of *Pseudomonas aeruginosa* reports up to 48% ceftolozane-tazobactam susceptibility and is associated with a combined resistance towards piperacillin-tazobactam, meropenem, imipenem, and ceftazidime [55,56].

Jorgensen et al. treated MDR and XDR phenotypes (95.8% & 37.7%) of *P. aeruginosa* isolates, respectively. The most common infection source was the respiratory tract (62.9%). High-dose of cefolozone/tazobactam was used in 71.2% of patients with a respiratory tract infection (RTI) overall but in only 39.6% of patients with an RTI who required C/T renal dose adjustment. In the primary efficacy population (n = 226), clinical failure and 30-day mortality occurred in 85 (37.6%) and 39 (17.3%) patients, respectively. New C/T MDR *P. aeruginosa* resistance was detected in 3 of 31 patients (9.7%) with follow-up cultures [57].

A high clinical success rate of 83.2% has been recorded using ceftolozane-tazobactam in the treatment of various types of *Pseudomonas aeruginosa* infections, of which 50.5% were XDR and 78.2% were resistant to at least one carbapenem [58]. Ceftolozane-tazobactam represents a good option for the treatment of susceptible MDR/XDR *Pseudomonas aeruginosa* infections, and it may be a first-line option in CR *Pseudomonas aeruginosa* infections, according to recent European guidelines [59,60].

*Ceftazidime-avibactam* is a combination of a well-known anti-pseudomonal third-generation cephalosporin with a new (non-β-lactam) β-lactamase inhibitor. It acts both (i) through ceftazidime, whose activity is expressed by linking to PBPs of the Gram-negative aerobic pathogens, including MDR or XDR strains of *Pseudomonas aeruginosa,* and (ii) through the ability of avibactam to overcome any kind of Ambler class of β-lactamases (type A (ESBL, KPC), C (AmpC cephalosporinases), and partially class D carbapenemase such as OXA-48 in *K. pneumoniae*) but not metallo-β-lactamases [61,62]. Tumbarello et al. performed a retrospective multicenter observational study on the use and outcomes of ceftazidime-avibactam therapy for infections caused by KPC *Klebsiella pneumoniae* strains. The cohort included 577 adults experiencing bacteremic (n = 391) or non-bacteremic infections. All were given treatment with ceftazidime-avibactam monotherapy (n = 165 patients) or in combination with other active antibiotics (n = 412 patients). The 30-day crude mortality was recorded at 25% (146/577). No difference in mortality was recorded between patients treated with ceftazidime-avibactam alone and those that were given combination regimens (26.1% vs. 25.0%, *p* = 0.79). In the logistic regression analysis, (i) the occurrence of a septic shock at the moment of infection onset (*p* = 0.002), (ii) the presence of neutropenia (*p* < 0.001), (iii) the increasing severity of sepsis severity (SOFA score ≥ 8) (*p* = 0.01), (iv) the experience of pneumonia (*p* = 0.04), and (vi) the necessity of ceftazidime-avibactam dose adjustment for the impairment of renal function (*p* = 0.01) negatively affected the patient’s mortality. On the other hand, better survival was associated with the prolonged infusion of ceftazidime-avibactam (*p* = 0.006) [63].

Tumbarello et el. conducted a retrospective observational study reviewing 138 cases of infections caused by KPC *Klebsiella pneumoniae* in adults who started ceftazidime-avibactam rescue therapy following a previous empiric treatment (median, 7 days) based on other antimicrobials. Ceftazidime-avibactam was given in combination with ≥1 active antimicrobial in 109 (78.9%) cases. The recorded 30-day mortality since infection onset was 34.1% for all the patients, while the 104 bacteremic KPC *Klebsiella pneumoniae* episodes showed a significantly higher survival than those of a matched cohort in which KPC *Klebsiella pneumoniae* B underwent therapy with drugs other than ceftazidime-avibactam (62.5% vs. 44.2%, *p* = 0.005). Multivariate analysis of the 204 KPC *Klebsiella pneumoniae* bacteremic episodes identified septic shock, neutropenia, Charlson comorbidity index ≥3, and recent mechanical ventilation as independent predictors of mortality, whereas treatment with ceftazidime-avibactam was the sole independent predictor for survival [64]. In a recent study, 3269 Enterobacteriaceae were consecutively isolated from critically ill patients experiencing community or nosocomial pneumonia. The most susceptible agents were ceftazidime-avibactam (99.9%), amikacin (98.7%), meropenem (97.4%), and tigecycline (94.6%); however, only ceftazidime-avibactam and tigecycline showed good activity (≥90% susceptible) against CR isolates (97.5% and 92.4%, respectively). In another study, the most active agents against MDR Enterobacteriaceae were ceftazidime-avibactam (99.2%) and amikacin (90.9%), if compared to ceftolozane-tazobactam (53.8%) and meropenem (78.1%). Moreover, among ESBL-producing Enterobacteriaceae, the susceptibility rates to ceftazidime-avibactam, meropenem, and ceftolozane-tazobactam were 100.0%, 84.1%, and 98.9%, respectively [65].

A recent review suggests the use of ceftazidime-avibactam for the treatment of microorganisms with reduced susceptibility (CP Enterobacteriaceae, CR, MDR, and XDR *Pseudomonas aeruginosa)* [66]. The INFORM database reveals that the susceptibility of *Pseudomonas aeruginosa* to ceftazidime-avibactam usually ranges from 88.7% to 93.2% in the four geographical areas [57]. In Western Europe, ceftazidime-avibactam maintains the best activity (87.8%) against *Pseudomonas aeruginosa* as compared to BL-BLI antipseudomonal drugs [67]. Moreover, ceftazidime-avibactam is related to high (87.8%) rates of clinical response in the treatment of serious infections caused by MDR and XDR GN and *Pseudomonas aeruginosa* isolates (33/41; 80.5% *Pseudomonas aeruginosa* infections) [68]. Lower (64%) ceftazidime-avibactam susceptibility rates have been recorded among pneumonia patients caused by DTR *Pseudomonas aeruginosa* [57]. A susceptibility of 71.7% to ceftazidime-avibactam has been reported among *Pseudomonas aeruginosa* that are MBL negative, while such a percentage decreases (19.1%) in MBL positivity [69,70]. Ceftazidime-avibactam is suggested as a targeted and proper treatment in DTR *Pseudomonas aeruginosa* infections [71], in relation to high susceptibility rates [59,72]. Patients experiencing pneumonia and kidney failure in continuous renal replacement therapy showed the development of resistance in the course of treatment [73,74]. The antibacterial spectrum of ceftazidime-avibactam makes ceftazidime-avibactam one of the first options for the empirical treatment of MDR, XDR, and CR/CP GN. Mantzarlis et al. conducted a study comparing 41 patients receiving ceftazidime-avibactam (cases) to 36 patients that received other antibiotics (controls) to treat severe infections caused by CR Enterobacteriaceae. They noticed in cases (i) a significant improvement of the SOFA score on days 4 and 10; (ii) a higher microbiological eradication (94.3% vs. 67.7%, *p* = 0.021) and clinical cure (80.5% vs. 52.8%, *p* = 0.010) rate; and (iii) a higher 28-day survival (85.4% vs. 61.1%). In the logistic regression model, the ceftazidime-avibactam-containing regimen was an independent predictor of survival and clinical cure (OR = 5.575, *p* = 0.012, OR = 5.125, *p* = 0.004, respectively). The authors concluded that a ceftazidime-avibactam-containing regimen was more effective than other available antibiotic agents for the treatment of CRE infections in high-risk patients [75].

Meropenem-vaborbactam is a new antimicrobial characterized by the association of meropenem and a new cyclic boronic acid β-lactamase inhibitor with a high affinity for the residues of serine. It works by determining a covalent bond with the β-lactamases without generating hydrolysis [76]. It is working against ESBL, AmpC, and serine carbapenemases (including KPC) microorganisms, while it is not effective against Ambler class B or class D carbapenemases [77,78]. Lapuebla et al. showed that 79% of GN isolates were susceptible to meropenem, but this rate was not increased by adding vaborbactam [79], since meropenem resistance is mainly caused by mechanisms due to porin mutations and up-regulation of efflux pumps, not antagonized by vaborbactam [80]. The “in vitro MIC” is the same for both agents; however, vaborbactam may increase killing activity since some GN strains may contain an inducible β-lactamase inhibited by vaborbactam [81]. In a recent study considering patients experiencing pneumonia due to both GN (3.193 isolates) and Enterobacteriaceae (4.790 isolates) and treated with meropenem-vaborbactam, 89.5% of GN were susceptible, and the susceptibility rates for MDR (21.8%) and for XDR (13.8%) were 59.0% and 48.6%, respectively [82]. The SENTRY Antimicrobial Surveillance Program (2014–2019) showed that the total susceptibility in vitro of *Pseudomonas aeruginosa* strains causing HAP and/or VAP to meropenem-vaborbactam in Europe was 82.1%. In oriental Europe, despite the great diffusion of KPC, the sensitivity rate of meropenem-vaborbactam was higher (89.7%) than in Eastern Europe, where MBL and OXA carbapenemases are more prevalent. In critically ill patients, meropenem-vaborbactam was working against 73.2% of *Pseudomonas aeruginosa* isolates, while the sole meropenem was active for only 57% of isolates [61,82]. For all 27% MDR and XDR *Pseudomonas aeruginosa*, the recorded susceptibilities of meropenem-vaborbactam and sole meropenem were 41% and 13%, respectively [61,83,84].

Imipenem-cilastatin-relebactam is a new association between imipenem and relebactam, a strong non-β-lactam bicyclic diazabicyclooctane β-lactamase inhibitor with an additional piperidine ring as compared to avibactam. Imipenem-cilastatin-relebactam works against both Ambler class A β-lactamases, (ESBLs and KPCs) and class C β-lactamases (AmpCs). However, relebactam cannot make imipenem active against OXA-48 and Ambler class B MBLs (IMP, VIM, and NDM)-producing isolates [85,86,87,88,89]. Imipenem-cilastatin-relebactam is active against CR strains using a chemical mechanism based on impermeability resistance [90,91,92,93]. Relebactam may re-establish imipenem susceptibility in up to 75–92% of imipenem-non-susceptible isolates [94,95,96,97]. Imipenem-cilastatin-relebactam showed in vitro activity against 82.2% of MDR *Pseudomonas aeruginosa* and only 62.2% of DTR *Pseudomonas aeruginosa* [98]. *Pseudomonas aeruginosa* causing both IAI and UTI showed a high susceptibility to imipenem-cilastatin-relebactam (96.7% and 96.4%, respectively), while imipenem-non-susceptible and MDR *Pseudomonas aeruginosa* strains were isolated in 85% and 87.3% of cases, respectively [99,100]. The SMART European surveillance study showed susceptibility to imipenem-cilastatin relebactam in *Pseudomonas aeruginosa* isolates in 94.4% of IAIs and 93% of UTIs. On the other hand, only 74.4% of imipenem-non-susceptible and 79.8% of MDR isolates were recorded from IAIs and UTIs [101]. Moreover, 91% of *Pseudomonas aeruginosa* isolates from the respiratory tract of ICU patients showed susceptibility to imipenem-cilastatin-relebactam [102].

In vivo studies established that imipenem-cilastatin-relebactam was more efficacious against MDR *Pseudomonas aeruginosa* than imipenem alone [103,104]. The RESTORE IMI-1 trial compared imipenem-cilastatin-relebactam to imipenem associated with colistin for the treatment of imipenem-non-susceptible *Pseudomonas aeruginosa*. A lower 28-day overall mortality was recorded in the case of imipenem-cilastatin-relebactam (9.5% vs. 30%) together with a better clinical response (71% vs. 40%) [98]. Rebold et al. treated 21 patients experiencing infections (52% LTRI) caused mainly by *Pseudomonas aeruginosa* (16/21, 76%), nearly all MDR (15/16, 94%), using imipenem-cilastatin-relebactam. They recorded both a 33% overall mortality rate and a 62% positive clinical response rate.

Cefiderocol is a siderophore cephalosporin with activity against a wide spectrum of GNs, including resistant ones. It works by linking to ferric iron and permeating the bacterial membrane [105,106]. Cefidecorol may achieve 99.6% susceptibility against all isolates and 97.3% in the case of XDR isolates, far higher than imipenem-relebactam (73.0%), ceftazidime-avibactam (73.4%), and ceftolozane-tazobactam (72.3%). Cefiderocol retains (i) complete activity against imipenem-relebactam-resistant isolates and (ii) significant susceptibility rates for isolates resistant to ceftazidime-avibactam and ceftolozane-tazobactam (91.6% and 88.3%, respectively) [107]. The CREDIBLE-CR trial documented its efficacy in CR GN infections, while in the APEKS-NP one, cefiderocol showed non-inferiority when confronted with a high dose of meropenem given in continuous infusion and similar safety in treating pneumonia caused by GNB, including MDR strains [108]. Cefiderocol monotherapy showed higher rates of clinical response and microbiological eradication than the comparators, providing a significant benefit in MBL-producing CR *Pseudomonas aeruginosa* infections too. Cefiderocol may be used in the treatment of microorganisms producing IMP, NDM, and VIM enzymes [109]. DTR *Pseudomonas aeruginosa* strains were even higher than the newer BL-BLI combinations [110].

Plazomicin is a new aminoglycoside recently approved by the FDA for the management of both cUTI and pyelonephritis caused by Enterobacteriaceae producing class A, C, and D β-lactamases [111], although it is not more efficacious than the other aminoglycosides against *Pseudomonas aeruginosa* and *Acinetobacter baumannii* [112]. The future is reserved for a lot of drugs (see Table 3).

##### Treatment of Ambler Class B β-Lactamases or MBL

The frequency of infections caused by MBL-producing Enterobacteriaceae and *Pseudomonas aeruginosa* is increasing all over the world and is usually associated with high mortality rates (>30%). The global spread of MBL-producing GN is a cause for concern for public health. Moreover, class B1 β-lactamases, including VIM-MBLs, IMP-MBLs, and NDM-MBLs, mostly carried by Enterobacteriaceae and *Pseudomonas aeruginosa*, have now spread everywhere, developing multitudes of clinical variants too. No consensus or shared guidelines exist to inform the management of these infections. MBLs can hydrolyze all β-lactams except for aztreonam; however, many strains may co-produce serine-β-lactamases enzymes (e.g., AmpC, ESBLs) that could hydrolyze such monobactam; therefore, a robust β-lactamase inhibitor, such as avibactam, could be given as a partner drug. As a consequence, aztreonam remains active against only about 30% of these isolates. Avibactam is efficacious against class A, C, and some D β-lactamases, including the clinically important enzymes CTX-M, KPC-2, AmpC, and OXA-48. Ceftazidime cannot be inhibited by carbapenemases such as OXA-48. It has been observed that aztreonam associated with both vaborbactam and relebactam may increase activity against class A serine-β-lactamases (i.e., KPC-3), in comparison to avibactam too [111].

Aztreonam/avibactam: In vitro data collected on 2209 GN show the high antimicrobial activity (MIC ≤ 4 mg/L) of aztreonam in combination with avibactam against (i) 80% of MBL-producing Enterobacteriaceae, (ii) 85% of *Stenotrophomonas* spp., and (iii) only 6% of MBL-producing *Pseudomonas*. Taking into consideration 64 patients experiencing only B, mortality occurred in only 19% of patients treated with aztreonam + ceftazidime-avibactam [113]. The association of aztreonam and a β-lactam/β-lactamase inhibitor such as ceftazidime-avibactam (ceftazidime-avibactam) has synergistic in vitro and in vivo activity, even against pathogens co-producing metallo- and serine-β-lactamases [113]. A prospective observational study conducted in three hospitals in Italy and Greece included 102 patients with B due to MBL-producing Enterobacteriaceae who were treated either with ceftazidime-avibactam + aztreonam or combinations of other in vitro active antibiotics. In 82 cases, the infection was caused by NDM-producing strains and, in 20 cases, by VIM-producing strains. Death occurred in 19.2% of the ceftazidime-avibactam + aztreonam group vs. 44% in the other one [OR 0.4 95% (CI 0.3–0.8) *p* = 0.007]. Treatment with ceftazidime-avibactam + aztreonam was associated with lower (i) 30-day mortality (*p* = 0.01), (ii) clinical failure at day 14 (*p* = 0.002), and (iii) shorter length of hospital stay (*p* = 0.007) [114]. In a prospective observational study, Falcone et al. compared the mortality of 102 patients experiencing a B, mainly caused by *Klebsiella pneumoniae* (91.2%), with only NDM (80.4%) and VIM (19.6%) producing and undergoing therapy with ceftazidime-avibactam + aztreonam or another active antibiotic. The 30-day mortality rate was lower for the group treated with ceftazidime-avibactam + aztreonam (19.2% (n = 52)) vs. (44% (n = 50); *p* = 0.007)). The differences in severity of illness between groups were equalized by a matched propensity score analysis, corroborating that the association of ceftazidime-avibactam + aztreonam was related to lower mortality. A proper infection model study found that human-simulated dosing of ceftazidime-avibactam (2–0.5 g every 8 h) + aztreonam (2 g every 6 h) over 2 h was related to the greatest bacterial killing activity without the emergence of resistance over 7 days [115]. The addition of avibactam at 4 mg/L determined reductions in aztreonam MICs to 1–2 mg/L [115].

The addition of avibactam or aztreonam induced significant reductions in bacterial colonies in an experimental thigh infection developed by a neutropenic mouse against MBL-producing Enterobacteriaceae compared to aztreonam alone [116]. Enterobacteriaceae co-harboring an AmpC, such as bla-CMY, may be prone to developing aztreonam-avibactam resistance, mainly in MBL-producing *Escherichia coli* [117,118,119,120].

##### The Future and in Development Drugs for GN

β-lactam-diazabicyclooctane β-lactamase inhibitor combinations diazabicyclooctanes (DBOs) are a class of β-lactamase inhibitors that includes older DBOs (avibactam and relebactam) and newer DBOs (zidebactam, nacubactam, durlobactam, and ETX0282) [121].

Meropenem-nacubactam (FPI-1465) is a non-β-lactam β-lactamase inhibitor with in vitro activity against class A, C, and some class D β-lactamases [122]. It showed significant reductions in bacterial burden against seven meropenem-resistant *Pseudomonas aeruginosa* clinical isolates in a neutropenic murine lung infection model [123,123].

Sulbactam-durlobactam (ETX2514) is a β-lactam with activity against *A. baumannii* and is a β-lactamase inhibitor working against class A β-lactamases. Its activity as a single agent against *A. baumannii* is limited due to its hydrolysis by various β-lactamases of class D.

Durlobactam (ETX2514) is a β-lactamase inhibitor that inhibits class A, C, and D β-lactamases. It also presents β-lactam properties, inhibiting PBP2 and thus having activity against some Enterobacterales [124].

Cefpodoxime proxetil-ETX0282 (active compound ETX1317) [125] β-lactam-boronate β-lactamase inhibitor combinations.

Cefepime-taniborbactam (VNRX-5133). Taniborbactam is a boronic-acid-containing β-lactamase inhibitor active against class A, C, D, and even class B β-lactamases, such as VIM, NDM, SPM-1, and GIM-1 (but not IMP). Its inhibiting activity works (i) by producing hydrolysis because it creates a covalent bind with the residue site of serine (serine-β-lactamases); and (ii) by the interaction of the boron portion with the active zinc site, tightening the active site split.

QPX7728 is an additional boronic acid containing a β-lactamase inhibitor, working against class A ESBLs and KPCs, class B (NDM, VIM, IMP), class C, and class D (OXA-48) in *Acinetobacter baumannii*) [126,127,128,129].

Recently, a study showed that the activity of cefepime-taniborbactam against meropenem-resistant *Pseudomonas* spp. producing serine-β-lactamase was superior to meropenem-vaborbactam, imipenem-cilastatin-relebactam, and ceftolozane-tazobactam, which was comparable to ceftazidime-avibactam but not for MBL-producing microorganisms [129,130]. A phase 3 non-inferiority study is currently ongoing to compare cefepime-taniborbactam with meropenem in the treatment of cUTI, with the endpoint being to assess the effectiveness of both intermittent and continuous infusion (over 2 h) dosing [131]. It could cover most MBLs too [132].

*Cefepime-Zidebactam:* Cefepime combined with zidebactam, a non-β-lactam bicycloacyl hydrazide pharmacophore, has good in vitro activity against MDR *Pseudomonas aeruginosa.* This association of cefepime, an inhibitor of PBP3, and zidebactam, an inhibitor of PBP2, results in an augmented bactericidal effect. It covers ESBLs, class C, OXA-48-like, and MBL-carbapenemases, although zidebactam cannot inhibit the two latter enzymes [133]. Cefepime-zidebactam is also likely to maintain activity against the most highly elevated efflux group of *Pseudomonas aeruginosa.* A total of 97 *(94.5%)* out of 103 *Pseudomonas aeruginosa*-producing ESBLs or MBLs were inhibited by cefepime-zidebactam, whereas fewer than 15% were susceptible to any other antimicrobial agent [134]. Indeed, cefepime-zidebactam shows high efficacy on XDR phenotypes of GNB, including *Pseudomonas aeruginosa* too [135]. A phase 3, multicenter PRCT has been registered in order to assess the efficacy and safety of cefepime 2 g and zidebactam 1 g intravenous vs. meropenem for the treatment of cUTIs or acute pyelonephritis in adults caused by GNB, including *Pseudomonas aeruginosa* [136,137].

### 2.3. Antibiotic Duration Therapy in GN Pulmonary Infections

HAP and VAP are the most recurrently ICU-acquired infectious complications, whose rates range from 5 to 40% in relation to ICU setting, case mix, and diagnostic criteria. In cases of their etiology due to non-fermentative GN bacilli (NFGNB), *Pseudomonas aeruginosa*, *Acinetobacter* spp., and *Stenotrophomonas maltophilia*, mortality rate and ICU LOS may significantly increase. The estimated attributable related mortality is about 10%, higher in surgical ICU patients and in patients with high severity scores [138]. Both European and US guidelines recommend a duration of antimicrobial therapy not longer than 7 days in most patients [138,139].

Several RCTs compared 8 vs. 15 days of antibiotic treatment in ICU patients who had developed VAP. Overall, no benefit was demonstrated in terms of mortality in extending antibiotic therapy duration up to 15 days, but in primary infections caused by NFGNB, a higher percentage of patients in the 8-day group developed documented pulmonary infection recurrence compared to those in the 15-day group [140]. A study specifically focused on early-onset VAP (5 to 7 days from hospitalization) showed that an 8-day course of antibiotic therapy is safe for early-onset VAP in patients on mechanical ventilation [141]. In a retrospective study on NFGNB VAP in a surgical/trauma ICU, 17% of patients were treated with a 3–8-day course of antibiotics, whereas 83% received nine or more days. A higher recurrence rate in patients with NFGNB VAP who received shorter courses of antibiotic therapy could not be demonstrated. On the contrary, patients who received shorter courses tended to have lower rates of recurrence [142]. In patients experiencing suspected VAP undergoing mechanical ventilation with stable ventilator settings (PEEP ≤ 5 cm H_2_O and FiO_2_ ≤ 40%), a short antimicrobial treatment (1–3 days) was associated with outcomes similar to those of longer courses (>3 days) [143]. Systematic assessment of ventilator settings may help ICU physicians identify candidates for early antibiotic withdrawal [144].

Another study focusing on HAP compared the effects of a shorter (5–7 days) to a prolonged course (10–14 days) of antibiotics on clinical resolution, super-infection, 30-day, and 90-day all-cause mortality. Patients on prolonged antibiotic regimens displayed a higher rate of super-infection, while 30-day mortality was higher in the short-course group [145].

### 2.4. Antibiotic Therapy Duration in IAI

IAIs include a spectrum of disease processes with a broad range of morbidity and mortality. Two major types of IAIs can be distinguished: uncomplicated and complicated.

The optimal management of complicated IAIs, defined as infections of the peritoneal space, is strictly related to early surgical control of the source, adequate antimicrobial treatment, and aggressive fluid challenge [146]. The etiology of IAIs is mainly due to GN bacilli and anaerobes, specifically *Escherichia coli* and *Klebsiella pneumoniae* [143]. In IAIs, patient-related factors, such as type of IAI, local disease, and concomitant secondary B, heavily contribute to patient outcomes [147]. Conventionally, a 7–14-day duration of antibiotic therapy has been advocated in patients with IAIs, but recently a short course of antibiotic agents (3–5 days) was recommended post-operatively, after achieving source control [148]. Antimicrobial therapy duration in patients with IAI needs to be individualized. The clinical decision to carry on or withdraw antimicrobial therapy should be supported by microbiological and laboratory data, particularly in cases of persistent IAIs [149]. The effect of antibiotic duration on infectious complications after laparoscopic appendectomy for acute complicated appendicitis was investigated in a prospective study comparing a 3-day to a 5-day course of antibiotic treatment. A shorter duration of therapy had no significant effect on infectious complications [150]. In patients developing localized peritonitis, associated with low, mild, or moderate severity of illness but requiring surgical intervention, the use of a short course of antibiotic therapy based on ertapenem showed the same clinical and microbiological efficacy as those obtained by a 5-day standard treatment [151].

Patel et al. compared the clinical outcomes of ICU patients who received a shorter (<7 days) vs. an extended (>7 days) course of antibiotics for B secondary to IAI. The main outcomes were the recurrence of the abdominal infections as well as the infections of the surgical sites and wounds, secondary yeast infections, and in-hospital mortality. Among surgical ICU patients developing B secondary to IAI, the withdrawal of antimicrobial treatment within 7 days of source control was not followed by an increased incidence of recurrent IAI [152]. In the STOP-IT trial, patients with complicated IAI and adequate source control were randomly divided into two groups: The first one that received fixed-duration antibiotic therapy (median duration therapy was 4 days) and the second one in which administration continued 2 days after the resolution of the physiological abnormalities (maximum 10 days, with a median of 8 days). The main outcomes were the development of a surgical-site infection, the relapsing of IAI, or death within 30 days following the surgery for source control, related to the treatment group. Secondary outcomes included the duration of antimicrobial therapy and the rate of eventual infections. No significant differences were recorded between groups in the rates of the primary and secondary outcomes. In patients developing IAI undergoing a proper source-control procedure, the outcomes after a standard duration of the antibiotic therapy (4 days) were similar to those treated with a longer course of antibiotics (8 days) [153]. Hassinger et al. retrospectively studied patients enrolled in the STOP-IT trial and found corticosteroid use, APACHE II score, HA origin of the infection, or a colonic source of the IAI as risk factors associated with treatment failure. The duration of treatment did not affect patients’ outcomes [154]. Another study investigated the impact of the STOP-IT trial on antibiotic usage and patient outcomes. The following endpoints were evaluated: Development of SSI, IAA, or fascial dehiscence; re-admission to the hospital within 30 days; and 30-day mortality. After source control, a short (four days or less) vs. a long (five days or more) course of antimicrobial treatment was compared. No differences were recorded between the two groups in terms of LOS, SSI, IAA, re-admission, death, composite outcome, or total costs. In the DURAPOP trial, the efficacy and safety of 8-day vs. 15-day antibiotic therapy in critically ill patients with post-operative IAI were compared. The main endpoint was the number of antibiotic-free days between randomization (day 8) and day 28. Treatments did not differ in terms of secondary outcomes, which were death, ICU and hospital LOS, the occurrence of MDR microorganisms, and the re-operation rate after a 45-day follow-up. Moreover, short-course antibiotic therapy in critically ill ICU patients experiencing post-operative IAI reduces antibiotic exposure, while prolongation of treatment until day 15 is not followed by any clinical benefit [155].

### 2.5. PK and PD in the Treatment of GN Infections

Having some understanding of PK/PD is important for clinicians when prescribing drugs, particularly in cases where there is a risk of the development of antimicrobial resistance [156]. Most antibiotics are usually given at standard dosing regimens, which do not take into account pathophysiologic and/or iatrogenic factors that are likely to affect the drug PK in “complicated” patients (Table 4). This scenario may be further complicated by the concomitant presence of factors with opposite effects on the PK of antibiotics, as in the case of critically ill patients with chronic or acute renal insufficiency, a transitory septic hyperdynamic phase leading to augmented renal drug clearance [157,158], or elderly patients with clinically relevant hypoalbuminemia needing treatment with highly effective antibiotics [158,159]. Since the MIC values reflect the power of the antibiotics, the best way to make a classification of such drugs is by the PK/PD relationship because it describes the drug exposure necessary to secure optimal drug effectiveness.

#### 2.5.1. Time-Dependent Antimicrobials

β-lactams are the classical time-dependent antibiotics. For these antimicrobials, maintaining TDM above the MIC (T>MIC) of the pathogen for a portion of the dosing interval has been shown to best predict microbiologic efficacy [158]. Indeed, T>MIC values of 40–70% have been suggested as targets for maximal bactericidal effect [91,92,93,94,95,96]. However, more recently, higher PK/PD ratios have been proposed to reach clinical cure. Indeed, some authors have set β-lactams cutoffs of T/MIC > 100%, >50–70%, or more ambitious targets of T/5xMIC > 50–70% [158]. The best way to maximize the PK/PD characteristics of time-dependent antibiotics is through increased frequency of drug administration or prolonged or continuous infusion regimens. The same trend was confirmed when comparing short (0.5 h) vs. prolonged (3 h) infusion regimens of 1 g meropenem every 8 h against CP *Klebsiella pneumoniae* Isolates with MICs of 4–8 mg/L [160,161].

#### 2.5.2. Concentration-Dependent Antimicrobials

This classification includes “pure” concentration-dependent antibiotics (such as aminoglycosides and daptomycin) best characterized by their peak-to-MIC ratio (Cmax/MIC). For these antibiotics, strategies aimed at maximizing the quantity of drug concentrations to be pursued, or concentration-dependent antibiotics with time-dependence (such as fluoroquinolones, glycopeptides, and oxazolidinones) characterized by AUC/MIC ratio, for which strategies aimed at maximizing the amount of drug exposure to be adopted for Cmax/MIC antibiotics, once-daily doses should be considered because these regimes have the highest chance to reach ideal PK/PD targets. Indeed, studies with gentamicin and amikacin have consistently shown that once-daily regimens were associated with a higher probability of reaching Cmax/MIC > 8 and better clinical responses in the treatment of GN pneumonia [162]. Similarly, specific PK/PD targets have been recently proposed for AUC/MIC antibiotics. For instance, levofloxacin AUC/MIC ratios > 125 have been associated with clinical and microbiological cure in critically ill patients, whereas several studies have consistently demonstrated that a target AUC/MIC > 400 is desired to obtain optimal vancomycin efficacy [158,159].

#### 2.5.3. Therapeutic Drug Monitoring (TDM)

TDM can be defined as the measurement of drug concentrations in an easily accessible biological matrix to be used for the eventual adjustment of the drug dosage based on PK principles. This approach is usually adopted in patients treated with narrow therapeutic index drugs, although it is extendable to all drugs [158,159].

The TDM of antibiotics has some peculiarities. Hence, the minimum therapeutic efficacy threshold, at variance with the large majority of other drugs, should rely not on specific drug concentrations but rather on PK/PD targets, as in the case of aminoglycosides, fluoroquinolones, glycopeptides, etc. The development of drug toxicity is eventually related to systemic drug overexposure, and not to PD [158,159].

#### 2.5.4. Antibiotic Interactions: Synergisms and Antagonisms

The progressive emergence of antibiotic resistance, the limited therapeutic options for the treatment of MDR-XDR-PDR microorganisms, and the lack of new molecules with novel mechanisms of action have prompted many physicians to test different antimicrobial combination therapies. From a theoretical viewpoint, the combination of two antibiotics may result in (a) a *cumulative antimicrobial effect,* which simply represents the sum of the two antimicrobials acting together; (b) a *synergism* between the two drugs, where the combined activity is greater than the sum of the individual antimicrobial activities; and (c) an *antagonistic interaction*, where the inhibitory effect of two antibiotics on bacterial growth is smaller than expected from their individual effects [162,163,164]. Despite the solid rationale for combining some antibiotics, most of the studies dealing with this therapeutic strategy provided disappointing results, with multiple combinations (such as vancomycin + daptomycin or linezolid, or the addition of gentamycin or rifampicin to either vancomycin or daptomycin) showing conflicting and inconclusive findings, with most of the combinations documenting no therapeutic advantages [163,164] or more toxicity compared with single treatments [165].

## 3. Materials and Methods

A literature search was performed through MEDLINE, PubMed, and Google Scholar, considering the period of time from 1 January 2015, up to 1 December 2022. The matches of the keywords were:(a)antibiotic duration therapy + MDR Gram-negative infection = 33 manuscripts(b)combination therapy + MDR Gram-negative infection = 156 manuscripts(c)monotherapy + MDR Gram-negative infection = 68(d)antibiotic duration therapy + XDR Gram-negative infection = 12 manuscripts(e)combination therapy + XDR Gram-negative infection = 60 manuscripts(f)monotherapy + XDR Gram-negative infection = 15 manuscripts(g)antibiotic duration therapy + PDR Gram-negative infection = 5 manuscripts(h)combination therapy + PDR Gram-negative infection = 15 manuscripts(i)monotherapy + PDR Gram-negative infection = 5 manuscripts

After removing duplicates, we found a total of 190 manuscripts.

Our selection criteria included articles in which the diagnosis of a Gram-negative infection was confirmed and that also reported data on antibiotic therapy duration. Articles published in the English language were selected, and any cited references were reviewed to identify relevant literature that included randomized clinical trials, prospective and retrospective studies, and case series that met our selection criteria. We excluded articles involving infections in children, pregnant women, and non-hospitalized patients.

## 4. Discussion

### 4.1. Treatment Duration: Short Courses or Long Courses? Combination or Monotherapy?

PRCTs focusing on the duration of therapy for severe and systemic infections have shown that treatment length can be reduced to 1 week or less without impairing patient outcomes [166,167]. Singer’s and Corona’s approach is based on using a 5–7-day SCAb short-course antibiotic therapy [43]. A total of 49 (80%) of 60 ICU-acquired B patients received SCAb monotherapy, associated with a recovery rate of 79%, 31.3% overall mortality, and 10.4% infection-associated mortality. Based on these data, the authors concluded that monotherapy is likely to provide satisfactory clinical responses. The same authors made a comparison with data collected through 2013, and the new study showed that short-course monotherapy was applied in 65.7% of episodes (73.5% in 2000). As with the 2000 cohort, the rate of antibiotic resistance, B breakthrough, and relapse remained low [168]. Havey et al. performed a meta-analysis of 13 studies reporting on 227 patients with B treated with “shorter” or “longer” durations of therapy. Among B patients undergoing shorter (5–7 days) vs. longer (7–21 days) antibiotic courses, no significant difference was assessed in terms of rates of clinical and microbiological response as well as survival [46]. Havey et al. performed a retrospective cohort study recruiting 100 critically ill patients developing a B: the median recorded duration of antibiotic therapy was 11 days, even if widely variable (4.5 to 17 days); indeed, clinical outcomes were similar between those receiving shorter and longer treatment [45].

The positive benefits of combination therapy for MDR/XDR GN infections remain controversial; however, the efficacy of specific antibiotic regimes has been investigated. Ceftolozane-tazobactam/fosfomycin association therapy was reported to be synergistic [169]. Ceftolozane-tazobactam shows no activity against CR *Pseudomonas aeruginosa* strains or MBL production, whereas ceftolozane-tazobactam and fosfomycin show in vitro synergy, leading to a decrease in ceftolozane-tazobactam MIC, even in cases of elevated MICs for both drugs [170]. Avibactam is important because it is active against *Pseudomonas* cephalosporinase and class A carbapenemases. In particular, the efficacy and synergy of the ceftazidime-avibactam/fosfomycin combination may lead to a significant decrease in 2 log colony-forming units against MDR/XDR GN. The combination of the two drugs may be considered a viable alternative in MBL-negative isolates [171]. The synergy of the association between fosfomycin and ceftazidime-avibactam was assessed by a small cohort of MDR *Pseudomonas aeruginosa*, half of which showed a high ceftazidime-avibactam MIC. Their combination caused a reduction “in vitro” in ceftazidime-avibactam MIC in 61.9% of strains [172].

In isolates with carbapenem resistance, the analysis of the association with fosfomycin and non-susceptible empirical antibiotics yielded in vitro synergy data in over 25% of all tested fosfomycin antibiotic combinations. The restoration of susceptibility is generally seen mostly in the case of cephalosporin + β-lactamase inhibitor combinations (i.e., fosfomycin/ceftazidime-avibactam 71.4%, fosfomycin/ceftolozane-tazobactam 68.8%) [173]. Following on this evidence, the proposed algorithms for the targeted treatment of MDR/XDR VAP due to *Pseudomonas aeruginosa* recommend fosfomycin combination therapy as a suitable and proper treatment and optimizing dosage option, just in the case of carbapenemase-producing MBL-negative *Pseudomonas aeruginosa* [174]. The need for a combined strategy with fosfomycin for the treatment of DTR *Pseudomonas aeruginosa* arose recently, particularly in relation to its in vitro bactericidal activity and selectivity of membrane channels [174]. Moreover, ceftolozane-tazobactam revealed a synergy in its association with colistin and in a triple regimen with fosfomycin in the treatment of infections caused by MDR *Pseudomonas aeruginosa* [169]. The synergy of ceftolozane-tazobactam in association with aztreonam is likely to decrease MIC in MDR-GN and *Pseudomonas aeruginosa* isolates [127]. The 2-fold reduction in ceftazidime-avibactam MIC in the case of its combination with amikacin, aztreonam, colistin, and meropenem has been found in a cohort of MDR Gram-negative patients, of whom half had *Pseudomonas aeruginosa* infections [170]. CR *Pseudomonas aeruginosa* and GN are in the “critical priority” group for which new antibiotics are urgently required. IDSA guidelines suggest ceftolozane-tazobactam, ceftazidime-avibactam, and imipenem-cilastatin-relebactam as monotherapy to be chosen as the main treatment option for infections outside of the urinary tract caused by DTR *Pseudomonas aeruginosa,* or GN. For serious infections due to MDR, XDR, or GN, it is still lacking high-quality data informing the decision in terms of using combinations or monotherapy, still today one of the most controversial questions in terms of antimicrobial strategy, even though the potential benefits of an empiric combination therapy are likely to increase the likelihood that at least one agent of the two is active, particularly in patients with a high risk of resistant strains, decreasing the risk of selection of a resistant sub-population, especially when the microbial burden is high.

Guidelines report no value in continuing combination therapy once in vitro susceptibility is confirmed [70,172]. SCAb may generally be considered for courses ≤ 5–7 days in length, whereas longer ones are ≥10–14 days. However, generally, SCAb may be a proper option (i) to reduce the risk of selection of resistance; (ii) in cases of quick onset of antibiotic action; (iii) for optimal penetration of the drug into the infection site; (iv) for antibiotics working towards non-dividing microorganisms; (v) for antibiotic action not affected by adverse infection; and (vi) no foreign (i.e., joint prosthesis, cardiac valve) body presents evidence of both abscess and any kind of immunodeficiency [173,174]. Infected prosthetic bodies (i.e., cardiac valve, vascular, and orthopedic prostheses) are not susceptible to short-course antibiotic monotherapy if not early removed, since a biofilm may form and encase the bacteria, decreasing the antibiotic penetration into the microorganisms. Since bacteria encased in biofilms have a reduced rate of both multiplication and metabolic activity, the targets for the action of some antibiotics diminish. Abscesses are not amenable to short antibiotic courses, either. Low pH and oxygen tension, typically present in a mature, developed abscess, may limit the activity of some antibiotics. Moreover, bacteria in biofilms are often in a non-dividing state, so antibiotic targets are not available because of their dependence on bacterial multiplication. The antibiotic chosen to treat SCAb should warrant a low propensity to induce resistance and easy penetration into tissues, thus being water-soluble, preferably of a lower molecular weight, and not tightly bound to serum proteins. Prolonged courses (>10–14 days) of intravenous antibiotics have been recommended for SAB because of concerns that infective endocarditis, deep infective foci, or metastatic infections might be present but not diagnosed [175]. A systematic review on the length of treatment of GNB secondary to UTI reported that the current limited evidence may corroborate that SCAb are as effective at obtaining clinical recovery microbiological eradication as longer courses [176]. Uno et al. performed a study suggesting that acute cholangitis with Gram-negative bacillary bacteremia can be treated safely with a shorter antimicrobial treatment duration of <14 days [166].

Indications for management of intra-vascular catheter-related BSI due to GNB recommend 7–14 days of therapy in the absence of complications based on the consensus opinion of experts [174,175,176,177]. Concern about infection recurrence and overall survival are the main factors contributing to prolonged antimicrobial therapy. To appease the adverse effects of longer antibiotic schemes, studies have investigated the clinical outcomes of patients with GNB associated with catheter-related infections in the urinary tract, abdominal, and pulmonary systems. Furthermore, these studies consider both stable and ICU patients [175,176,177,178].

Several studies compared short and prolonged antibiotic therapy for GNB BSI in hemodynamically stable patients in terms of mortality and B recurrence. The same survival rates were found in the two treatment groups [175,176,177,178].

Furthermore, the odds of relapse of B and CD infections were also similar. A protective effect of short-course antibiotic therapy on the occurrence of MDR GN bacteria was recorded [175,176,177,178]. A study including 54 patients with GNB catheter-related infections showed that adequate antimicrobial therapy given for 7 days may be as safe and effective as longer courses in episodes of GN catheter-related B infections once the CVC has been removed [179].

Fabre et al. showed that patients affected by *Pseudomonas aeruginosa* B and undergoing shorter courses were discharged from hospital 4 days earlier than those who were treated with longer courses of intravenous therapy [180]. In a retrospective cohort study, Nelson et al. examined the effectiveness, in terms of the risk of treatment failure, of short (7–10 days) and long (>10 days) courses of antibiotic therapy for uncomplicated Gram-negative BSI. The risk of treatment failure was significantly higher with a short course (7–10 days) of antimicrobial therapy compared to a long one (>10 days), with failure rates of 0% for patients treated for 12 days [181]. The same results were found by Giannella et al. [182] and Corona et al. in a multicenter multinational prospective observational study in which antimicrobial duration therapy was not found to be a predictor of death [183]. A retrospective cohort study was conducted by Chotiprasitsakul et al. at three medical centers, including patients with Enterobacteriaceae bacteremia. The median duration of treatment was 8 days (IQR 7–9) in the short-course group and 15 days (IQR, 13–15) in the prolonged one. No differences were found in mortality between the two groups (adjusted hazard ratio (aHR): 1.00; 95% confidence interval (CI): 0.62–1.63).

Short-course antibiotic therapy had a protective effect on the emergence of MDR-GN microorganisms (OR, 0.59; 95% CI, 0.32–1.09; *p* = 0.09). Both short courses of antibiotic therapy and prolonged ones are related to similar clinical outcomes in the case of Enterobacteriaceae B and may prevent subsequent MDR-GN bacteria [183]. Moreover, Ruiz-Ruigomez et al. showed that an appropriate antibiotic therapy given for less than 7 days may be as safe and effective as longer courses in episodes of GN catheter-related infections if the CVC has been removed [184].

### 4.2. What May Influence (Short vs. Long/Mono vs. Combination) Antibiotic Prescribing Practice?

Unfortunately, no PRCT exists to inform best practice in terms of optimal type and duration of treatment of GN infections, particularly in relation to the progressive spread of MDR, XDR, DTR, and PDR bacteria causing deep-seated infections with a high degree of severity of illness. The major fear is supposed to be the “timely” selection of microorganisms resistant to the new and future antimicrobials. Only observational audits have been reported, along with indications of key opinion leaders. The literature reports many approaches, widely varying according to geographic distribution and local therapeutic policies and strategies. This major weakness of these variable approaches could be the increasing global antibiotic resistance due to the global trend of long-course combination therapies that indeed do not protect from treatment failure and relapsing infections. A questionnaire survey was performed among a total of 254 ICUs from 34 countries [59]. A small percentage (3–15%) of ICUs using SCAb (≤5–7 days) for primary B was recorded. Authors found that the direct input from microbiologists or infectious disease specialists had a statistically significant *inverse impact* on the duration of antibiotic therapy: “*the greater was the microbiologist/infectious* diseases specialist input the shorter was the duration” [167]. Bornard et al. reported that the intervention of three visits in a week by an infectious diseases specialist to evaluate the ongoing antibiotic therapies, interactive teaching courses, and daily contact with a microbiologist triggered an improvement of the applied antibiotic strategies or schemes [173]: Reductions in extended-spectrum penicillins and carbapenems, but an increase in narrow-spectrum penicillins. An antimicrobial stewardship program may help to “rationalise a systematic approach to the use of antimicrobial agents to achieve optimal outcomes” [173]. However, short-course therapy was significantly associated with relapse (7.9% vs. 0%; *p* = 0.036). Corona and Singer et al.’s daily practice [43,44] is to treat bacteremia with a short course (5 days) of monotherapy, unless otherwise indicated.

Table 5 shows the main characteristics of patients’ and microorganisms’ risk stratification in terms of the proper commencement of short or longer courses of therapy. Intrinsic and specific antibiotic characteristics may help predict the length of an efficacious therapy. This is in terms of the availability of bactericidal vs. bacteriostatic antibiotics, synergistic combinations, or the capability of surgically removing the source. A published prospective observational study identified that neither any type of antibiotic policy nor any kind of interaction between the effectiveness of therapy and the type of approach affects survival.

For MDR, Gram-negative III/IV generation cephalosporins or BLIBL in association with aminoglycosides or quinolones are the most common choices. For ESBL-producing or AmpC-producing Gram-negative patients, early therapy with carbapenems seems to be the right choice if compared to BLIBL, whereas cefepime and the new drug ceftolozane-tazobactam make sense in the modern view of carbapenem sparing. For CP Enterobacteriaceae (or XDR and PDR Gram-negative), the antimicrobial regimens are supposed to be based on the newly developed drugs in Table 6. Actually, the reports of associations between antimicrobials are as varied as the involved carbapenemases. Experimental antibiotic regimens are mainly reported in observational studies. Combination therapy regimes may also be based on the synergies and associations shown in Table 5. PK/PD is important for clinicians when prescribing drugs to reduce the development of antimicrobial resistance. For this reason, in recent years, it has become clear that the concept of “fast pharmacology” is integrated with “fast microbiology” in order to be as targeted and adequate as possible with antimicrobial therapy, using information from both TDM and MIC. The PK of an antibiotic may have to be matched with its PD. The best PD marker for these drugs is MIC. For time-dependent antibiotics, maintaining drug concentrations above the MIC (T>MIC) of the pathogen for a portion of the dosing interval has been shown to best predict microbiological efficacy. For concentration-dependent antibiotics, the peak-to-MIC ratio, or AUC/MIC ratio, is the target to be considered. TDM adds efficacy and adequateness to antimicrobial regimens. Attention is to be paid to synergism or antagonism between the specific drugs in terms of optimal duration, which may generally be considered courses ≤ 5–7 days, whereas longer ones ≥ 10–14 days. However, very few datasets can be found to support the former or latter approach. The reported basic requirements for determining the length of courses are (i) a totally susceptible pathogen; (ii) a low or no risk of selection of antimicrobial resistance; (iii) bactericidal drugs given in an adequate regimen; (iii) a rapid onset of antibiotic action; (iv) a good tissue penetration percentage of the antibiotic into the infection site; (v) an antibiotic active against non-dividing bacteria; (vi) antibiotic activity not affected by adverse infection; (vii) no foreign body or prosthetic material, no abscess organization, or no signs of humoral or cellular immunodeficiency. Clinicians should remember that all treatment decisions are dynamic, therefore requiring frequent reassessment depending on patient response. Certainly, the timely selection of an appropriate and adequate antibiotic regimen should, intuitively, impact patient outcome rather than its duration. Table 6 is the synthesis of the main approaches and antibiotic choices to treat GN severe infections in relation to antimicrobial resistance. SCAb could be expanded both in terms of indications and in terms of geographical extension. Why not challenge our current fear that makes us treat severe infections (intra-vascular, abdominal sepsis, chest infection) or severely ill patients (i.e., immunocompromised, SOT, HIV positive) with a long course of antibiotics. Mathematicians did the same with the five Euclidean postulates, and the universe was better understood. Why do we remain steady in the contraposition between the Ptolemaic vision (long courses) and the Copernican one (short courses) without giving clear hints? Short or long is better is not a syntactically correct sentence, as it can never be proved or disproved within the same system as the first Kurt Godell’s incompleteness theorem sentence.

**Table 6 antibiotics-12-01262-t006:** Therapeutic strategies: synthesis.

Type of Microorganism	Presence of Risk Factor in Table 7	Absence of Risk Factor in Table 7
MDR Gram-negative	Combination therapy Reports: suggesting short courses	Combination therapy Reports: suggesting short courses
ESBL-producing Gram-negative AmpC-producing Gram-negative	Combination (when feasible) therapy No clear indication in terms of duration	Combination (when feasible) therapy No clear indication in terms of duration
CP-producing Gram-negative	Combination (when feasible) therapy No clear indication in terms of duration	Combination (when feasible) therapy No clear indication in terms of duration

**Table 7 antibiotics-12-01262-t007:** Antibiotic treatment for Gram-positive and GN B (see Conclusions and Expert Commentary).

Microorganisms	Gold Standard	Combination with	Second Line	Combination with	New Drugs
MDR	Third-generation cephalosporins BLIBL Quinolones	+ Quinolones + aminoglycosides	Quinolones III–IV generation Cephalosporins Cefepime/cefmetazolo Tigecycline ^#^ Colistin ^#^	+ aminoglycosides	Ceftazidime-avibactam Ceftolozane-tazobactam
ESBL-positive Gram-negative	Carbapenems	+ Fosfomicin + aminoglycosides + tigecycline	Ceftolozane/tazobactam	+ Fosfomicin + aminoglycosides + tigecycline	See Table 4
AmpC-BL-producing	Carbapenems		Ceftolozane/tazobactam	+ Fosfomicin + aminoglycosides	See Table 4
XDR, PDR, DTR CP—Gram-negative	Ceftazidime-avibactam Imipenem-cilastatin-relebactam Meropenem-vaborbactam Cefidecorol meropenem)	+ fosfomicin + aminoglycosides + colistin	Double carbapenem regimen (doripenem + meropenem) (ertapenem +	+ Fosfomicin + aminoglycosides + colistin	Cefepime-taniborbactam Cefepime-zidebactam See Table 4
Amber class B β-lactamases (NDL, VIM)	Aztreonam-avibactam Ceftazidime-avibactam Meropenem-Vaborbactam	+ fosfomicin ^§^ + aztreonam	Ceftazidime-avibactam Meropenem-vaborbactam	+ Aztreonam	β-lactam–diazabicyclooctane β-lactamase inhibitor combinations β-lactam–boronate β-lactamase inhibitor Combinations. See Table 4

(^#^) in case of HR/(^§^) not proved.

## 5. Conclusions and Taking Home Message and Commentary

✓B infections are prevalent and caused by MDR microorganisms. Moreover, the end of the last century has progressively seen an upsurge in XDR and PDR rates, responsible for concern towards the theoretical end of the antibiotic era.✓The timely commencement of adequate and appropriate antibiotic therapy has a strong impact on patients’ outcomes.✓A strategy of short-course therapy may represent—in the contemporary context—A reasonable and logical choice for treating B, considering the stratification of risk shown in Table 7. Indeed, it may present a few advantages in terms of (i) reduced risk of central catheter-associated complications, including bloodstream complications; (ii) antibiotic resistance development, antibiotic-associated organ toxicity, and drug interactions; (iii) decreased costs and increased efficacy; and (c) improved convenience and treatment compliance. Antibiotic characteristics may help to predict the length of effective therapy, such as the availability of bactericidal vs. bacteriostatic drugs or synergistic combinations and the capacity to control the source of infection.✓Clinicians have to consider that such a strategy is to be considered for restricted indications, as shown in Table 5, Table 6 and Table 7.✓Clinicians have to remember that all treatment decisions are dynamic, requiring frequent—at least daily—reassessment in relation to the patient’s clinical response, and what is important is the timely commencement of an adequate and appropriate antibiotic therapy whose duration is eventually to be decided.

## Figures and Tables

**Table 1 antibiotics-12-01262-t001:** Epidemiology of nosocomial and community-acquired GN infections.

	Prevalence	
Microorganisms	Nosocomial (*)	Community Acquired (§)
*Pseudomonas* spp.	8–25.3%	
*Pseudomonas aeruginosa*	24–27.8%	2.8–5.9%
*Acinetobacter* spp.	2.2–41.5%	0.3%
*Acinetobacter baumanii* calcoaceticus complex	32.7–36%	
*Acinetobacter lwoffii*	4.9%	
*Stenotrophomonas* spp.	19.9%	
*Stenotrophomonas maltophilia*	1.4–8.9%	0.5%
*Burkholderia pseudomallei*	2.70%	0.7–4.8%
*Escherichia coli*	24–44%	29.8–63.5%
*Klebsiella* spp.		3.4–7.6%
*Klebsiella pneumoniae*	22.5–40%	3.7–17.6%
*Klebsiella oxytocha*	1.4%	1.1%
*Enterobacter* spp.	5%	1.8–1.9%
*Citrobacter* spp.	1.3–1.7%	/
*Proteus mirabilis*	1.5–7%	1.8–2.6%
*Serratia marcescens*	1.9–11.1%	0.5–1.4%
*Neisserira meningitidis*	/	0.1–4.7%
*Salmonella typhi*	/	29.6–51.1%
*Shigella* spp.	/	0.1%
*Haemophylus influenzae*	/	9.8%

Legend: (*) The shown prevalence data for B refer to studies only considering GN-B, whereas the others (§) refer to epidemiological studies and surveys considering the total number of bacteremia, both due to GN and Gram-positive microorganisms [5,6,7,8].

**Table 2 antibiotics-12-01262-t002:** Rate of multi (M), extensively (X), and pan (P) drug-resistant (DR) Gram-negative microorganisms.

Microorganisms	MDR	XDR	PDR	CP-R	ESBL Producer
*Pseudomonas* spp.				17.4–26.6%	
*Pseudomonas aeruginosa*	15.7–39.6%	8.9–23%%	2–5.9%	4.5–35%	17–90%
*Klebiella* spp.			Case-reports-3.9/4.3%	6.7–8.5%	2.5–100.0% 18%
*Klebsiella pneumoniae*	2–44.6%	7–26.6%		<1–50%	
*Klebsiella oxytoca*	2%	3%		<1%	
*Escherichia coli*	2–50%	7.7–11.5%	5–6.2%	0.1–12.1%	0
*Acinetobacter* spp.	31–70/90%		12.29%	56.6–77%	
*Acinetobacter calcoaceticus* baumanii complex		15–40%	14.2–20%		90–99%
*Enterobacter* spp.	2–28.7%	16%	2.4–6.7%	0.9–1.7%	
*Enterobacter cloacae*					6–22%
*Peoteus mirabilis*	3.8%	15.7%	0.7–1.7	4.6%	
*Stenotrophomonas maltophilia*	17–30.5%	39.4%	/	/	0
*Serratia* spp.	30%	4.7–6.2%	1.5%	0.3–1.5%	0
*Citrobacter* spp.	34.5%	1%	<1%	0.5–1%	0

Legend: According to the review, a wide variation exists in the literature in terms of the use of the definitions of XD-R and PD-R, and sometimes these definitions overlap [5,6,7,8].

**Table 3 antibiotics-12-01262-t003:** Future antimicrobial agents with activity against Gram-positive and Gram-negative microorganisms.

Antimicrobial Agents	Class	Mechanism of Actions	Susceptibility
Ceftaroline-avibactam		Inhibitors of serine-β-lactamases	
Imipenem-cilastatin-relebactam (formally MK-7655)	Serine lactamase inhibitor reversible, covalent non-β-lactam, β-lactamase inhibitor	Inhibitors of serine-β-lactamases	CP-EnB
Eravacycline	A novel synthetic fluorocycline	With a potency two to four times greater than tigecycline	CRE
Aztreonam WCK4234 Ceftazidime + WO2013/030735 meropenem WCK 5153 Aztreonam + FPI-1465 Ceftazidime	Diazabicyclooctanones + (monobactam or ceftazidime or meropenem)		Oxacillinase-producing strains of *Acinetobacter baumannii*. Antibacterial activity against *Pseudomonas aeruginosa* and *Escherichia coli* ESBLs and class A, B, and D carbapenemases.
Meropenem + CB618.40/CB-618	Diazabicyclooctanones + meropenem		Enterobacteriaceae expressing the KPC-2, KPC-3, FOX-5, OXA-48, SHV-11, SHV-27, and/or TEM-1 β-lactamases
Meropenem-RPX700 Biapenem-RPX7009	Boronic acid β-lactamases inhibitor + carbapenems		Many class A and C serine-β-lactamases
Benzo(b)thiophene-2-boronic acid	Boronic acid β-lactamases inhibitor + ceftazidime		Many class A and C serine-β-lactamases
Cefepime + AA1101	Cefepime + novel sulfones/clavam		
Imipenem + MG96077	Phosphonates + imipenem		Reducing >90% of the MICs of *Pseudomonas aeruginosa* and *Klebsiella pneumoniae* at 4 mg/L resistant to imipenem.
Imipenem + MK-8712 Aztreonam + siderophore-monobactam Meropenem Ceftazidime + Syn2190 Cefpirome	Carbapenems + monobactams		Circumvent certain β-lactamases inhibit certain AmpC β-lactamase activity against *Acinetobacter* spp., which include those with some blaOXAs, *P. aeruginosa*, *Burkholderia* spp., and Enterobacteriaceae lower MIC values to the susceptible range against *P. aeruginosa* demonstrated activity in mouse systemic and urinary tract infection models using *P. aeruginosa*
S-649266 3′-thiobenzoyl-cephalosporin FSI-1671 and FSI-1686	Novel siderophore cephalosporin Novel 3′-thiobenzoyl-cephalosporin + meropenem Novel carbapenems		A novel catechol-substituted siderophore cephalosporin demonstrated activity against *P. aeruginosa*, *S. maltophilia*, *K. pneumoniae*, and *A. baumannii* Combined with meropenem, may have activity against *P. aeruginosa*, *S. maltophilia*, and *Chryseobacterium meningosepticum* demonstrated inhibitory activity against class A, B, C, and D β-lactamases. Active against MDR *Acinetobacter baumannii*, Enterobacteriaceae, and some *Pseudomonas aeruginosa* active against carbapenem-resistant *Acinetobacter baumannii,* **Klebsiella pneumoniae*,* and *Pseudomonas aeruginosa*
Bisthiazolidine ME 1071 (a maleic acid) Biapenem-ME1071	Metallo-β-lactamase-specific inhibitors		Active against *Acinetobacter baumannii*, *Klebsiella pneumoniae*, and *Providencia rettgeri* producing blaNDM-1. The combination with ceftazidime increased its susceptibility to *Pseudomonas aeruginosa* expressing blaIMP and blaVIM. Decreased MICs for Enterobacteriaceae with blaIMP and blaVIM, but not blaNDM.

**Table 4 antibiotics-12-01262-t004:** Patho-physiological changes and their effects on antibiotic PK.

Changes	Effect on Drug PK
Impaired absorption	Reduced bioavailability of orally administered antibiotics
Hypoalbuminemia	Increased Vd and clearance of highly bound (>80%) antibiotics with increased risk of failing to attain PK/PD targets
Obesity	Increased Vd and changes in hepatic metabolism and renal excretion, especially in hydrophilic antimicrobials (e.g., β-lactams, vancomycin). Lack of specific dosing recommendations for antibiotics in obese patients
Renal failure	Impaired clearance of hydrophilic antibiotics with augmented risk of joining over-therapeutic plasma concentrations and undergoing drug-related toxicity (β-lactams, aminoglycosides)
Hyperfiltration	Increased clearance of hydrophilic antibiotics with increased risk of reaching sub-therapeutic plasma concentrations (β-lactams)
Modified fluid balance (increased capillary leakage)	Increasing extracellular fluid volume causing the augmentation of the drug Vd and reducing plasma drug concentrations, particularly in case of the hydrophilic antibiotics with low Vd (aminoglycosides, β-lactams)
Clearance related to CVVH (DF)	CRRT (continuous renal replacement therapy) is associated with increased Vd and clearance of hydrophilic antibiotics

**Table 5 antibiotics-12-01262-t005:** Risk stratification for decision towards short or longer courses of antibiotic therapy.

Factors Influencing Decisions
Microorganism characteristics Metastatic infection potential Attachment behaviors and biofilm formation Kinetics of growth Susceptibility pattern of microorganisms: MDR, XDR, PDR
Patient characteristics Immune status (i.e., HIV, autoimmune diseases) Comorbidities (i.e., cirrhosis, IDDM) Foreign material Place of acquisition (community vs. nosocomial)
Infection characteristics Duration of infection site Source of infection (e.g., immunologically privileged, inadequate blood flow) Severity of illness (e.g., shock, severe sepsis) Poorly penetrated sites (e.g., the central nervous system, the prostate, poorly vascularized tissue, and endovascular vegetation)
Effects of therapy efficacy/failure Response to therapy, Antimicrobial intrinsic characteristics (bactericidal/bacteriostatic) bioavailability of the drug at infection site (monotherapy vs. combination therapy) Source control or eradication Surgical vs. non-surgical (e.g., removal of catheter, drainage)

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
