# Peer review of "Antibiotic Therapy Strategies for Treating Gram-Negative Severe Infections in the Critically Ill: A Narrative Review"

_antibiotics, 2023, doi:10.3390/antibiotics12081262_

Round 1
Reviewer 1 Report
I read with interest the present review entitled Antibiotic Therapy Strategies For Treating Gram Negative Severe Infections In The Critically Ills: A Narrative Review, by Corona et al. The topic is hot and the authors present useful data for those who deal with infections management. I find the review quite extensive but the authors should focus on their main questions which are duration and type (combination or monotherapy) of therapy. I find that there is a lot of information not relevant to their main questions. I have some suggestions to make to the authors.
Comments
1.Αbstract-conclusion: It is too general. Everybody agrees that decisions should not be taken mechanistically but on an individualized manner. I would expect from the authors to conclude what their review adds to the current knowledge as a conclusion.
2.Methods: While the authors provide in the abstract, a brief description of their methodology in identifying the literature they have worked on, their description in the text follows result which is confusing. Please remove it just after introduction.
Results:
3. My main remark is the organization of the material. I would expect to see a text organized in a manner that responds to the main two questions. Thus, a first part to deal with the duration of therapy in various studies in regard with a) location of infection b) clinical severity of infection and c) microbiology (MDR or Not). One or more tables relevant to these points would be helpful
4.The authors present al lot of information which may be interesting but not directly linked to the main questions. It is difficult for the reader to link all the detailed presentation in the text. For example what is the point to present many data on novel antibiotics or on pk/pd?
Or what do a table like No 3 in such a review that focuses on duration/type of therapy. Do the authors suggest that different antibiotic categories require different duration of treatment? If yes they could add a section d (see comment 3) to what I have suggested before and they could discuss all relevant points.
5. A second part in respect of the combination or not therapy, organized as the first part would be helpful (see comment 3).
6. Discussion should be brief and should discuss the main questions. Especially, the authors could underline a) the quality of studies b) the populations studied and c) the need or not, and the focus of future studies in the field
7. Some numbers of references in the text are underlined others not? Please check for type errors , for example in discussion “Same Authors…”
8. I think Matzarlis et al, present useful data related to MDR infections. The authors might consider also include these data if they think it is relevant.
Author Response
1.Αbstract-conclusion: It is too general. Everybody agrees that decisions should not be taken mechanistically but on an individualized manner. I would expect from the authors to conclude what their review adds to the current knowledge as a conclusion.
CHANGED
2.Methods: While the authors provide in the abstract, a brief description of their methodology in identifying the literature they have worked on, their description in the text follows result which is confusing. Please remove it just after introduction.
CHANGES
Results:
- My main remark is the organization of the material. I would expect to see a text organized in a manner that responds to the main two questions. Thus, a first part to deal with the duration of therapy in various studies in regard with a) location of infection b) clinical severity of infection and c) microbiology (MDR or Not). One or more tables relevant to these points would be helpful
- CHANGED
4.The authors present al lot of information which may be interesting but not directly linked to the main questions. It is difficult for the reader to link all the detailed presentation in the text. For example what is the point to present many data on novel antibiotics or on pk/pd?
Or what do a table like No 3 in such a review that focuses on duration/type of therapy. Do the authors suggest that different antibiotic categories require different duration of treatment? If yes they could add a section d (see comment 3) to what I have suggested before and they could discuss all relevant points.
CHANGED
- A second part in respect of the combination or not therapy, organized as the first part would be helpful (see comment 3).
- MADE
- Discussion should be brief and should discuss the main questions. Especially, the authors could underline a) the quality of studies b) the populations studied and c) the need or not, and the focus of future studies in the field
CHANGED
- Some numbers of references in the text are underlined others not? Please check for type errors , for example in discussion “Same Authors…”
CHANGED
- I think Matzarlis et al, present useful data related to MDR infections. The authors might consider also include these data if they think it is relevant.
IF REFERENCE IS BETTER SPECIFIED I WILL BE MORE THAN HAPPY TO PUT IT INTO THE MANUSCRIPT. I HAVE FOUND A GREE AUTHOR "mantzaris" IS HE?

Reviewer 2 Report
the idea of the article is interesting and I can appreciate the work that went into making in happen, what I had trouble with is the fact that you are missing the methods section altogether in your article.
Also, section 2.2.1. is missing any text.
I also feel that some English language editing is required in order to make this an article worth publishing.
Author Response
the idea of the article is interesting and I can appreciate the work that went into making in happen, what I had trouble with is the fact that you are missing the methods section altogether in your article.
Also, section 2.2.1. is missing any text.
I also feel that some English language editing is required in order to make this an article worth publishing.
ALL CHANGES REQUESTED MADE

Reviewer 3 Report
This was a review of a manuscript entitled Antibiotic Therapy Strategies For Treating Gram Negative Severe Infections In The Critically Ills. The review was comprehensive, well-written, and may fill a knowledge gap. Nonetheless, a few minor issues must be addressed.
1. This page contains many Abbreviations. However, following the author's guidelines, abbreviations should be defined the first time they appear in each of three sections.
For example Page 5, Line 163
CPE “Please use the full name of Carbapenemase-producing Enterobacteriaceae (CPE)”
Page 5, Line 181 CRE “Please use the full name of Carbapenem-Resistant Enterobacteriaceae (CRE)”
2. Table 1: "Neisserira meningitidis" please correct the phrase.
3. The author must include the exact date of the literature search in the Method section.
Author Response
This was a review of a manuscript entitled Antibiotic Therapy Strategies For Treating Gram Negative Severe Infections In The Critically Ills. The review was comprehensive, well-written, and may fill a knowledge gap. Nonetheless, a few minor issues must be addressed.
- This page contains many Abbreviations. However, following the author's guidelines, abbreviations should be defined the first time they appear in each of three sections.
For example Page 5, Line 163
CPE “Please use the full name of Carbapenemase-producing Enterobacteriaceae (CPE)”
Page 5, Line 181 CRE “Please use the full name of Carbapenem-Resistant Enterobacteriaceae (CRE)”
- Table 1: "Neisserira meningitidis" please correct the phrase.
- The author must include the exact date of the literature search in the Method section.
ALL THE MODIFICATIONS REQUESTED HAVE BEEN MADE

Round 2
Reviewer 1 Report
The authors responded adequately to my comments.
Mantzarlis K. is a Greek author who has contributed with some papers on MDR infections. The authors could use his work as a reference if they think his work is relevant (I leave it to authors decision; this is my only and minor comment to offer.
Author Response
I add the following manuscript as asked.
Tsolaki V, Mantzarlis K, Mpakalis A, Malli E, Tsimpoukas F, Tsirogianni A, Papagiannitsis C, Zygoulis P, Papadonta ME, Petinaki E, Makris D, Zakynthinos E.
Ceftazidime-Avibactam To Treat Life-Threatening Infections by Carbapenem-Resistant Pathogens in Critically Ill Mechanically Ventilated Patients. Antimicrob Agents Chemother. 2020 Feb 21;64(3)
